# New Insights on the Nuclear Functions and Targeting of FAK in Cancer

**DOI:** 10.3390/ijms23041998

**Published:** 2022-02-11

**Authors:** Silvia Pomella, Matteo Cassandri, Maria Rita Braghini, Francesco Marampon, Anna Alisi, Rossella Rota

**Affiliations:** 1Department of Oncohematology, Bambino Gesù Children’s Hospital, IRCCS, 00146 Rome, Italy; silvia.pomella@opbg.net (S.P.); matteo.cassandri@opbg.net (M.C.); 2Department of Radiotherapy, Policlinico Umberto I, Sapienza University of Rome, 00185 Rome, Italy; francesco.marampon@uniroma1.it; 3Unit of Molecular Genetics of Complex Phenotypes, Bambino Gesù Children’s Hospital, IRCCS, 00146 Rome, Italy; mariarita.braghini@opbg.net

**Keywords:** FAK, adult cancers, pediatric cancers, targeted therapy, combination therapy, PROTACs, ATP-competitive inhibitors, allosteric inhibitors

## Abstract

Focal adhesion kinase (FAK) is a non-receptor tyrosine kinase over-expressed and activated in both adult and pediatric cancers, where it plays important roles in the regulation of pathogenesis and progression of the malignant phenotype. FAK exerts its functions in cancer by two different ways: a kinase activity in the cytoplasm, mainly dependent on the integrin signaling, and a scaffolding activity into the nucleus by networking with different gene expression regulators. For this reason, FAK has to be considered a target with high therapeutic values. Indeed, evidence suggests that FAK targeting could be effective, either alone or in combination, with other already available treatments. Here, we propose an overview of the novel insights about FAK’s structure and nuclear functions, with a special focus on the recent findings concerning the roles of this protein in cancer. Additionally, we provide a recent update on FAK inhibitors that are currently in clinical trials for patients with cancer, and discuss the challenge and future directions of drug-based anti-FAK targeted therapies.

## 1. Introduction

Focal adhesion kinase (FAK) is a non-receptor tyrosine kinase with a molecular weight of 125 kDa that contributes to the regulation of different cellular processes, including cell survival, proliferation, apoptosis, adhesion, migration and mechano-transduction (ref. [1] and reviewed in [2]). The FAK gene, also known as protein tyrosine kinase 2 (*PTK2*), maps onto the human chromosome 8q24.3 region and encodes for the FAK protein that is ubiquitously expressed in different types of cells, with a prevalent localization in the plasma membrane and cytoplasm. FAK was described for the first time in 1992 by Schaller et al. as a central protein associated to the sub-cellular structures named focal adhesions (FAs), which regulate signaling events in response to the extracellular matrix (ECM) [3]. Indeed, FAK was long considered only as the central transducer of extracellular signaling, the most important triggered by integrins or growth factors, through its association with several proteins located in close proximity to the focal contacts. Over the last ten years, in addition to this kinase-dependent function, an unexpected kinase-independent role in the scaffolding and regulation of gene transcription was also discovered for FAK (ref. [4] and reviewed in [5,6]).

Several studies have shown the ability of FAK in regulating angiogenesis, immune cell recruitment, ECM remodeling, epithelial--mesenchymal transformation and stem cell maintenance, thus supporting the crucial role of this protein in cancer development and progression (reviewed in [7,8]). As a matter of fact, the last two decades of research have provided evidence for a role of FAK in promoting different types of tumors, including the most common adult breast cancer and hepatocellular carcinoma (HCC), but also pediatric cancers such as Ewing sarcoma and rhabdomyosarcoma (ref. [9] and reviewed in [10,11,12]).

In the present review, we summarize the current knowledge on FAK structure and the novel insights into its scaffolding nuclear functions, mainly focusing on recent updates of its role in cancer.

We also give an overview of FAK inhibitors currently in clinical trials on patients with cancer and discuss the challenge and future directions of drug-based anti-FAK targeted therapies.

## 2. FAK Structure and Activation

FAK’s structure includes three main domains: an N-terminal FERM (four-point—1, ezrin, radixin and moesin—domain, a central kinase domain followed and a C-terminal FAT (focal adhesion targeting) domain (reviewed in [13]) that are separated by three proline-rich regions (PR1-3) working as binding sites for the Src homology (SH) 3 domain-containing proteins (Figure 1). The FERM domain contains nuclear export signals (NES) and nuclear localization signals (NLS), while another NES is located in the central kinase domain (reviewed in [14]). The FAT domain consists of a four-helical bundle, wherein the major role of FAK recruitment to FAs is by the interaction with other proteins, such as talin and paxillin (reviewed in [15]). The FAT domain also contains the Tyr925 phosphorylation site, which is relevant for its binding to the SH2 domain of Src kinase and the consequent activation thereof [16]. It has been suggested that, to be phosphorylated at Tyr925, the FERM domain should be unfolded (reviewed in [13]). The FAK/Src complex represents a platform for the docking and phosphorylation of additional signaling molecules, including the extracellular signal-regulated kinase 2 (ERK2)/mitogen-activated protein kinase (MAPK), which is a key regulator of cell proliferation and differentiation [17]. The central kinase domain includes the ATP binding site with the activation loop from residues 564–585 [18].

The activation of FAK occurs through auto-phosphorylation in the Tyr397 residue, which is located in a linker segment that connect the FERM with the kinase domain (Figure 1) [14]. Once this site is phosphorylated, the full catalytic activation of FAK is achieved by the binding of Src to the phosphorylated Tyr397 site and consequent phosphorylation of the FAK activation loop on Tyr576 and Tyr577 residues [14]. Once activated, FAK takes part in a number of different downstream signaling pathways involved in the regulation of cell survival, proliferation and motility. In particular, FAK promotes the recruitment of paxillin and talin, involved in the assembling and correct turnover of FAs, essential dynamic events for the control of cell motility and directional cell migration. FAK also plays an important role in the regulation of the actin cytoskeleton through interactions with several regulatory proteins connected to the activation of Rho family proteins. Moreover, FAK assures the transmission of survival signals through integrin-dependent adhesion and signaling, in a process called anoikis, in the absence of which the cells undergo apoptosis. Indeed, in adherent cells, FAK transmits signals blocking anoikis [19]. Furthermore, in response to stress signals or cell detachment from the extracellular matrix, FAK can enter in the nucleus via its NLS to regulate different intranuclear pathways [20].

## 3. Nuclear FAK Functions Related to Cancer

A number of evidences suggest that FAK can translocate into the nucleus in response to specific conditions such as the detachment of the cells from the ECM, the activation of stress signals and the inhibition of kinase activity (reviewed in [5]). In the nucleus, FAK (i) controls the functions of a variety of transcription factors (TFs), impacting gene regulation; (ii) acts as a scaffold for several oncogenic and tumor-suppressor proteins by stabilizing their complexes with different ubiquitin protein ligases E3, thus, promoting their turnover through enhanced ubiquitination and the consequent proteasomal degradation (reviewed in [5]) (Figure 2).

One of the earliest discovered kinase-independent scaffolding functions of FAK in the nucleus regards its ability to stabilize a p53-MDM2 (an ubiquitin E3 ligase) complex, leading to the tumor suppressor p53 poly-ubiquitination and subsequent degradation by the proteasome promoting survival of normal cells [20]. This feature plays a role during development, when p53 must be expressed at low levels to allow the proliferation of cell precursors [20]. Golubovskaya et al. previously discovered that FAK directly interacts with p53 through the N-terminal domain (see Figure 1), forming a complex localized in both the cytoplasm and nucleus [21]. The authors also showed that, in the human osteogenic sarcoma p53-null cell line SAOS-2, the inhibition of FAK, when p53 is re-expressed, resulted in cell death, suggesting that FAK was responsible for the blockade of p53-mediated apoptosis and transcriptional activity [21]. The enhancement of p53 degradation by FAK interaction with p53-MDM2 complex in the nuclear and perinuclear compartments may explain why the activity of p53, as a tumor suppressor, could be impaired in p53 wild-type (wt) tumors (such as neuroblastoma) despite its functional integrity [22]. In agreement, small molecule-inhibiting FAK-p53 interactions screened on colon cancer cells reduced cell viability, reactivating p53-dependent transcriptional activity and was synergic with conventional anti-cancer chemotherapeutics [23]. These findings demonstrated that, through the shuttling of the FAK-p53 complex between the two cellular compartments, FAK acts on p53 through distinct mechanisms by promoting its degradation and, in the meantime, by inhibiting p53-mediated gene transcription and anti-survival functions. Further, Serrels et al. demonstrated that nuclear FAK can also regulate cell-cycle progression by controlling the RUNX1-dependent expression of the insulin-like growth factor-binding protein 3 (IGFBP3) in skin squamous cell carcinoma [24]. Specifically, FAK into the nucleus recruits Sin3a, which represses the transcriptional activity of RUNX1, to a complex with RUNX1, thus inhibiting the expression of IGFBP3. The latter, when re-expressed in FAK-depleted cells, prevented proliferation and tumor growth in vivo [24]. Notably, RUNX1 is known to interact with an array of proteins, including kinases, chromatin remodeling enzymes and ubiquitin ligases, a number of which interact with FAK that, thus, can regulate RUNX1 activity in different manners [24].

Moreover, several studies suggest a scaffolding role of nuclear FAK in complexes that modulate chromatin accessibility. Luo et al. described, for the first time, a mechanism through which nuclear FAK interacted with methyl CpG-binding domain protein 2 (MBD2) in myogenic cells, shuttling to the nucleus where the MBD2-FAK complex altered heterochromatin reorganization, decreasing the MBD2 association with HDAC1 (histone deacetylase complex 1) and methyl CpG site at the myogenin promoter [25]. As a result, myogenin was then expressed, followed by muscle differentiation [25]. Both MBD2 and HDAC1 take part of the nucleosome remodeling and deacetylase (NuRD) complex, one of the major chromatin remodeling complexes mainly associated with gene silencing [26,27]. In line with this study, we analyzed the nuclear interactome of FAK and the components of FAK complexes in HCC by mass spectrometry and noticed that HDAC1 and HDAC2 emerged as novel FAK nuclear interactors [4]. Genetic or pharmacological suppression of FAK functions decreased the nuclear amount of HDAC1/2 proteins, reduced their activity and, so, determined an increase of lysine 27 acetylation on histone H3 (H3K27ac), potentially indicating activation of gene transcription [4]. In a previous work, we reported an inverse role for FAK in promoting the translocation of a target protein from the nucleus to the cytoplasm, thus, hampering its nuclear functions. Indeed, we showed that FAK interacted with EZH2, a histone methyltransferase that trimethylates histone H3 on lysine 27 (H3K27me3), repressing gene transcription [28], and that this interaction delocalized EZH2 from the nucleus blocking its nuclear functions [29]. As a result, the expression of genes directly repressed by EZH2, such as NOTCH2, a well-known EZH2 target in HCC [30], was restored [29]. Moreover, we demonstrated an additional layer of regulation by FAK of EZH2, since FAK depletion reduced HCC growth in vitro and in vivo by preventing the expression of EZH2 and other cancer-promoting genes, through the reduction of the binding of the TFs E2F2/3 to the EZH2 promoter, thus causing a consequent decrease of H3K27me3 levels [29]. Of note, in our subsequent works, this FAK–EZH2 complex has been shown to be crucial also for the β-catenin-driven hepatocarcinogenesis [31,32].

The ability of FAK in the regulation of chromatin accessibility and TFs binding to DNA is also discussed by several studies that describe the protein in an immunomodulatory context. GATA4 is another TF modulated by FAK scaffolding functions to regulate inflammation-induced gene expression. As a matter of fact, in cardiac endothelial cells in vivo, nuclear FAK interacts with CHIP (C terminus of Hsp70-interacting protein), an E3 ligase that polyubiquitinates GATA4, thus leading to GATA4 degradation and the consequent limitation of GATA4-induced VCAM-1 expression in response to tumor necrosis factor-α (TNF-α) [33]. This effect was triggered by the inhibition of FAK kinase, which promoted FAK nuclear shuttling and was independent from NF-kB [33]. VCAM-1 is an adhesion molecule that plays a major role in transendothelial migration of lymphocytes and cancer cells and, in agreement with this data, the pharmacological inhibition of FAK reduced VCAM-1 expression in tissues, in vivo, and impaired metastasis in a murine model of melanoma [34]. Furthermore, nuclear FAK has been be shown to be associated with chromatin and to interact with TFs, including the TBP-associated factor TAF9 and others transcriptional regulators reported or predicted to regulate the expression of the chemokine Ccl5 [35]. Then, the same authors showed that both IL33 and ST2 are transcriptionally regulated by nuclear FAK, confirming its role in regulating cytokine expression and tumor growth [36]. A recent study demonstrated that nuclear FAK regulates IL33 expression by controlling c-Jun binding at the IL-33 enhancer region via chromatin accessibility changes [37]. All these findings suggest that FAK may also control the transcription of chemokines, thus expanding the capacity of the FAK nuclear interactome to regulate the composition of the immunosuppressive tumor microenvironment. Lastly, nuclear FAK is required for angiogenesis. Indeed, a direct role for nuclear FAK in the transcriptional regulation of vascular endothelial growth-factor receptor 2, a central mediator of endothelial cell proliferation and migration in angiogenesis, has been recently reported [38].

Looking at all this evidence, nuclear FAK influences multiple cellular pathways by direct protein interactions, but the mechanisms that underly this probably non-catalytic aptitude of the protein still remain to be clarified.

## 4. FAK Inhibitors

The critical and important role of FAK in the cancer progression of a plethora of human tumors [11,12,39,40] led to the development of selective FAK-targeting small molecules.

FAK inhibitors can be divided into three different categories: (i) kinase domain inhibitors, (ii) allosteric inhibitors and (iii) proteolysis-targeting chimera (PROTACs).

(i) Among the kinase domain inhibitors, the ATP-competitive ones are the most investigated FAK inhibitors. The ATP-competitive molecules bind to the FAK–kinase domain, directly competing with ATP, thus, inhibiting FAK signal transduction activity and the activation of several FAK downstream pathways (reviewed in [41,42]). Being able to selectively bind to the FAK ATP-binding domain, they are considered the most promising molecules to be translated and applied in clinical practice. Indeed, all the compounds inhibiting FAK that have accessed clinical trials and are currently under evaluation in Phase I and II belong to this category [42] (Table 1). One of the major challenges in the development of FAK inhibitors is to achieve a high selectivity towards proline-rich tyrosine kinase 2 (PYK2), another member of the FAK family. PYK2 is a tissue-specific non-receptor tyrosine kinase encoded by the protein tyrosin kinase 2 beta (*PTK2B*) gene (on Chr. 8p21.2), sharing 78% homology with FAK at the ATP binding site and a similar multi-domain organization [43,44]. Additionally, PYK2 regulates FAs formation and has been shown upregulated during FAK signaling suppression, thus, compensating for the loss of FAK, potentially promoting resistance [45,46]. Moreover, similarly to FAK, PYK2 can translocate from the cytoplasm to the nucleus thanks to its inability to associate with talin, which reduces its localization on FAs [47]. Additional similarities have been demonstrated between FAK and PYK2 in the nucleus, where, also, PYK2 can form complexes with p53 and MDM2, promoting p53 degradation in normal and cancer cells [47], and it can bind MDB2 [25]. However, differently from FAK, PYK2 has a unique characteristic, translocating to the nucleus in response to Ca++ signals in neurons [48]. Overall, the differences between PYK2’s and FAK’s nuclear functions, and their importance in FAK’ signaling and inhibition, still remain to be clarified.

(ii) The allosteric inhibitors of FAK are non-ATP-competitive compounds still undergoing pre-clinical investigation. Allosteric inhibitors have been developed that bind an allosteric site within the kinase domain, but, differently from the ATP-binding site, and seem to be highly specific to FAK [49,50,51,52]. These inhibitors induce an inactive conformation of the kinase domain, hampering interactions with receptor tyrosine kinases (RTKs) or auto-phosphorylation at Tyr397.

Among the allosteric inhibitors are also those compounds that bind to non-kinase domains of FAK, such as FERM and FAT, acting on the scaffolding functions of the kinase by interrupting or avoiding protein–protein interactions (PPIs) between FAK domains and their associated proteins. FAK-MDM2, FAK-p53 and FAK-VEGFR3 PPIs have been intensely studied and specific allosteric compounds were discovered (reviewed in [53]).

(iii) PROTACs are new-generation compounds developed in the last years as inducers of protein degradation (ref. [54] and reviewed in [55]). PROTACs are heterobifunctional molecules able to hijack the ubiquitin–proteasome system (UPS) to degrade specific target proteins by concomitantly binding an E3 ubiquitin ligase, among which Von Hippel Lindau (VHL) or Cereblon (CRBN), and the selected protein. Conversely to traditional small molecules, PROTACs can target “undruggable” proteins lacking relevant binding sites but having pockets with small affinity for compounds that can be designed or found by screening, such as transcription factors or nuclear proteins. PROTACs against kinases have been developed and tested in pre-clinical settings [56,57]. Notably, one of the major benefits of PROTACs is their selectivity due to the specific interaction between the E3 ligase and the protein target.

Recently, a number of PROTACs against FAK that include a binder for the VHL E3 ligase have been developed [58,59,60]. PROTAC-3 was developed based on the ATP-competitive FAK inhibitor Defactinib (Table 1) fusing to the E3 ubiquitin ligase VHL, and has been shown to be more effective in inhibiting the activation of FAK and FAK-dependent cell migration and invasion in breast cancer cells in vitro [58]. FC-11 is a FAK PROTAC molecule obtained by the fusion of the VS-6062 ATP-competitive inhibitor (Table 1) with the E3 ubiquitin ligase CRBN. In vivo experiments showed FAK degradation in reproductive mouse tissues associated with a more than 90% reduction of total and phosphorylated (Tyr397) protein levels [59]. Very recently, the GSK215 FAK PROTAC has been developed from the ATP-competitive FAK inhibitor VS-4718 (Table 1) [60]. The efficiency of GSK215 was testified to by the marked reduction of FAK levels in liver tissues, in vivo. This pharmacologic study clearly shows that FAK inhibition and FAK degradation have different effects, since GSK215 inhibited in vitro cell motility and 3D growth while VS-4718 was unable to do so.

## 5. FAK Inhibitors in Clinical Trials

In Table 1 we summarize FAK inhibitors that are being evaluated in clinical trials as single agents or in combination. Of note, all the clinical trials using FAK inhibitors are focused on adult patients.

GSK2256098 (GTPL7939) is a potent, highly selective and reversible ATP-competitive FAK inhibitor (dissociation constant (Ki) = 0.4 nM). GSK2256098 inhibits FAK Y397 phosphorylation in several cancer cells. The drug treatment affects AKT and ERK downstream pathways, thus, impairing cell viability and anchorage -independent growth and inducing caspase-mediated apoptosis in L3.6P1 pancreatic ductal adenocarcinoma cells [61]. HepG2 HCC cells treated with GSK2256098 decreased phosphorylation levels of PI3K, AKT, STAT3 and JNK correlated with an anti-proliferative effect [62]. Preclinical data showed that GSK2256098 treatment reduces microvessel density and cellular proliferation and induces apoptosis more efficiently in PTEN-mutated than in PTEN wt uterine cancer cells [63]. Glioblastoma cells were found to be among the most sensitive to GSK2256098 in a screening of 95 cancer cell lines [64]. In agreement, in vivo experiments demonstrated that, in a human glioblastoma xenografted model, GSK2256098 treatment induced a time- and dose-dependent inhibition of FAK by reducing its phosphorylation [64]. To date, GSK2256098 is under investigation in six clinical trials, of which four are in Phase I (completed) and two in Phase II (one recruiting and one active but not recruiting) (Table 1). Two Phase I studies have enrolled healthy volunteers to evaluate and determine safety and biodistribution of GSK2256098 (NCT00996671, NCT02551653). A Phase I study, in the United Kingdom, on 62 patients with advanced solid tumors showed encouraging results on the acceptable safety profile and activity of GSK2256098 in mesothelioma patients (NCT01138033) [65]. A Phase Ib trial of GSK2256098 in combination with trametinib, a MEK/MAPK inhibitor, was conducted on 34 patients (of which 21 had malignant mesothelioma). Results from the study suggested that co-administration with GSK2256098 increases the trametinib uptake (NCT01938443) [66]. Moreover, two Phase II clinical trials have been activated to evaluate: (1) GSK225098 in combination with trametinib in advanced pancreatic cancer (NCT02428270; active, not recruiting) and (2) GSK225098 in combination with vismodegib, a hedgehog inhibitor, in intracranial and recurrent meningioma (NCT02523014; recruiting).

VS-4718 (PND-1186) is a potent, reversible and selective FAK inhibitor (Ki = 1.5 nM). VS-4718 IC50 dose reduced FAK Y397 phosphorylation in breast carcinoma cells, leading to tumor growth arrest and apoptosis induction [67]. A panel of 47 human cancer cell lines were tested for sensitivity to VS-4718, among which were renal cancer, thyroid cancer, ovarian cancer, breast carcinoma, melanoma, mesothelioma and non-small-cell lung cancer cell lines [68]. Overall, data from this study suggested that the absence of the Merlin tumor suppressor correlates with high sensitivity to VS-4718 treatment in malignant pleural mesothelioma in vitro and in vivo. VS-4718 showed potent inhibition activity in vitro in a pediatric preclinical testing program (PPTP) and excellent tolerance in vivo [69]. Moreover, VS-4718 can act as a competitive substrate for ABCB1 and ABCB2, thus affecting the activity of these transporters and leading to the intracellular accumulation and increased efficacy of small molecules [70]. Recently, transcriptomic analysis performed on a uveal melanoma cell line treated with VS-4718 revealed that the treatment downregulates genes stimulated by KRAS, EGFR and cytokines, such as *IL-21* and *IL-15*. VS-4718 repressed also the expression of genes downregulated by JAK2, p53 and BMI. Interestingly the authors observed a down-regulation of YAP signature genes due to a strong reduction of YAP nuclear localization after VS-4718 treatment [71]. In a preclinical study unrelated to cancer, VS-4718 was used to inhibit FAK in proliferating vascular smooth cells (vsmc) to show that inactive FAK enters to the nucleus where it forms a complex with Skp2, an E3-ubiquitin ligase, and CDH1, an activator for APC/C E3 ligase complex, to promote their degradation [72]. The results were the increase of the two cyclin-dependent kinase inhibitors, p21 and p27, and the consequent blockade of vsmc proliferation [72]. VS-4718 was investigated in three Phase I clinical trials (Table 1) but all of them were either terminated with no available results (NCT01849744, NCT02651727) or withdrawn (NCT02215629).

Defactinib (PF04554878, VS-6063) is a potent dual and reversible ATP-competitive inhibitor of FAK and PYK2 (Ki = 0.6 nM, both) [73,74]. Preclinical studies revealed that Defactinib reduced FAK Y397 phosphorylation in a dose-dependent manner, and combinatorial treatment with paclitaxel, a chemotherapy drug, reduced cell proliferation and induced apoptosis in ovarian cancer cells [75]. Moreover, Defactinib can overcome the in vitro paclitaxel-resistance mediated by the DNA- and RNA-binding proteins YB-1 [75]. Defactinib induces dissociation of PI3K from FAK in esophageal squamous cell carcinoma, thus resulting in impaired AKT signaling and in the transcriptional downregulation of several oncogenes such as *SOX2*, *MYC*, *EGFR*, *MET*, *MDM2* and *TGFBR2*, thus reducing tumor growth and metastatic ability [76]. Human malignant mesothelioma (MM) cells overexpressing calreticulin, a Ca2+-binding protein critical for MM cell survival in vitro, show increased nuclear FAK and resistance to Defactinib in vitro [77]. The co-treatment with Defactinib and docetaxel, another chemotherapy drug, impaired the proliferation of castration-resistant prostate cancer cells in vitro and in vivo [78]. Defactinib is under evaluation in 21 clinical trials: 9 in Phase I (2 terminated, 5 completed, 1 recruiting and 1 withdrawn), 2 in Phase I/II (both recruiting) and 10 in Phase II (2 terminated, 1 completed, 6 recruiting and 1 active but not recruiting) (Table 1). Phase I studies established the acceptable safety, tolerability, pharmacokinetics profile and clinical activity in 9 patients (NCT01943292) [74] and 46 patients with advanced solid tumors (mostly colorectal, ovarian or pancreatic cancer) (NCT00787033) [79]. In the Phase II trial NCT01951690, Defactinib monotherapy showed modest clinical activity in heavily pretreated KRAS mutant non-small cell lung carcinoma (NSCLC) patients [80]. A Phase II study, involving 344 patients affected by malignant pleural mesothelioma, demonstrated that Defactinib treatment after first line chemotherapy did not improve either progression-free survival (PFS) or overall survival (OS) (NCT01870609) [81].

VS-6062 (PF00562271) is a potent dual and reversible ATP-competitive inhibitor of FAK and PYK2 (Ki = 1.5 nM and Ki = 14 nM, respectively) [82]. VS-6062 potently reduces FAK Y397 phosphorylation in epidermal squamous cell carcinoma [82] and Ewing sarcoma cell lines [83], resulting in the repression of downstream pathways. Preclinical studies demonstrated that co-treatment with VS-6062 and Sunitinib, a multi-targeted RTK inhibitor (RTKi), strongly inhibits angiogenesis and proliferation in liver and epithelial ovarian cancers [84,85]. Furthermore, VS-6062 treatment impairs T cell proliferation, adhesion to ICAM-1 (intercellular adhesion molecule-1) and interactions with antigen-presenting cells [86]. VS-6062 treatment reduced FAK activation and consequently SRC and BCAR1 phosphorylation, inhibiting cell growth and inducing apoptosis in liposarcoma cells [87]. VS-6062 was evaluated in a Phase I clinical trial (Table 1) in which 99 patients with advanced solid tumors were enrolled. Results from the trial showed a safety profile of VS-6062 and a time-dose dependent non-linear absorption, distribution, bioavailability, metabolism and excretion (NCT00666926; completed) [88].

CEP-37440 is a potent dual and reversible ATP-competitive inhibitor of FAK and ALK (Ki = 2.3 nM and Ki = 120 nM, respectively). In vitro treatment with CEP-37440 reduced the cell proliferation of anaplastic large-cell lymphoma cells [89]. CEP-37440 was able to completely inhibit the proliferation of FC-IBC02 breast cancer cells in vitro, affecting the transcriptional expression of genes related to apoptosis, interferon signaling and cytokines such as *IFI27*, *IFI6*, *IFI35*, *IRF7*, *CCL5*, *IL32*, *IL23A*, *OAS2*, *OAS3*, *OAS1*, *MX1*, *ISG15*, *BIK* and *KDR* [90]. Furthermore, CEP-37440 showed efficacy in breast cancer preclinical models both in vitro and in vivo [90]. It also exhibited good oral ADME (absorption, distribution, metabolism and excretion) properties, high bioavailability in several animal species (mouse, rat and monkey) and excellent activities in in vivo models of ALK- and FAK-positive tumors [89,90]. Furthermore, CEP-37440 is a brain-penetrant drug [91]. CEP-37440 was evaluated and successfully completed Phase I clinical trials (Table 1). Thirty-two patients with advanced or metastatic solid tumors were enrolled to determine the maximum tolerated dose (MTD), safety and tolerability of oral CEP-37440 (NCT01922752), but the results are not available.

BI-853520 is a selective and potent FAK inhibitor that binds the FAK kinase region, blocking ATP access [73]. It inhibits FAK Y397 phosphorylation in prostate cancer cell lines with an IC50 of 1 nM [92]. Furthermore, it has been demonstrated that it reduces tumorsphere formation and in vivo orthotopic malignant pleural mesothelioma growth [93]. In addition, recent studies reported that BI-853520 has high specificity for FAK in breast cancer cells [94]. Indeed, it represses FAK activity through the inhibition of Y397 autophosphorylation, while in FAK’s homologue PYK2 phosphorylation was unaffected [94]. RNA-seq analysis performed on BI-853520-treated 4T1 breast cancer cells xenografted in mice revealed the downregulation of genes involved in proliferation and cell cycle progression, such as *CDK1* and *CDK4*, and the upregulation of genes involved in T-cell differentiation and proliferation, cytokine production and leukocyte activation [94]. Currently, BI-853520 effects are under investigations in three different Phase I clinical trials, of which two are completed (NCT01905111, NCT01335269) and one is recruiting (NCT04109456). NCT01905111 clinical trial assessed the safety, tolerability, MTD and preliminary data on antitumor effects of BI-853520 monotherapy in a cohort of 21 Taiwanese and Japanese patients affected by various advanced or metastatic tumors. The results showed that BI-853520 has an acceptable safety profile and potential antitumor effects [95]. BI-853520 MTD and antitumor efficacy was assessed also in a Phase I clinical trial on 96 patients affected by advanced and metastatic non-hematologic tumors (NCT01335269). The trial was completed and showed that BI 853520 has an acceptable safety profile and modest antitumor activity at a MTD of 200 mg in the selected patients’ cohort [96]. Finally, a recruiting, Phase Ib clinical trial (NCT04109456) is aimed to investigate safety, tolerability, pharmacokinetics and anti-tumor effects in metastatic melanoma patients.

## 6. Future Directions and Concluding Remarks

FAK signaling represents a convergence node of key cellular processes regulated by integrins and growth factor receptors to drive numerous cell functions. It was found de-regulated in several tumor types, including both adult and pediatric forms, suggesting its targeting is helpful for the treatment of patients with cancer. So, over the last years, a big effort is ongoing to discover novel FAK inhibitors with growing selectivity that could be translated to the clinic. However, although FAK inhibitors as monotherapies showed good tolerability and pharmacokinetic profiles, they seem to mainly have cytostatic effects and limited efficacy in extending progression-free survival [66,73]. In addition, targeted therapy with single agents often results in drug resistance through different mechanisms, orienting the clinical approach to combinatorial treatments.

Based on this evidence, several clinical studies are evaluating FAK inhibitors in combination with other agents, as reported in Table 1. Indeed, anti-FAK compounds are under clinical study in combination with RTKi, such as the MEK1/2 inhibitors Trametinib and Cobimetinib and the RAF/MEK inhibitor VS-6766 (reviewed in [97]). The combination with the chemotherapeutics Paclitaxel alone or with Carboplatin is under evaluation. Moreover, since nuclear FAK has been shown to remodel chromatin to lead the transcription of pro-inflammatory cytokines, thus contributing to immunoevasion, evaluation of the combination with the immune checkpoint inhibitor Pembrolizumab is also ongoing (reviewed in [97]).

Small molecules targeting FAK PPIs could be pharmacological tools to block the non-kinase activity of FAK, avoiding the selectivity issues of ATP-competitive compounds (reviewed in [53]). However, the development of FAK scaffolding inhibitors still retains the limitations related to stoichiometric drug binding and occupation of the binding site needed to modulate protein function, thus, resulting in a weak potency [60].

An additional aspect that should not be underestimated is that FAK and PYK2 can have opposite, redundant or synergistic effects, depending on the cell type and condition (reviewed in [98]). Therefore, the choice of dual FAK/PYK2 inhibitors in the clinical setting should be carefully evaluated and corroborated by supportive preclinical studies.

The development of PROTACs seems to have a great clinical potentiality allowing the degradation of a specific protein even when it is mutated during treatment. It is noteworthy that PROTACs against FAK have the potentiality to affect both kinase dependent and independent (scaffolding) functions. Moreover, FAK can be activated by compensatory signaling pathways such as other RTKs that trans-phosphorylate Tyr397 overcoming the inhibition of FAK activation by conventional kinase-domain inhibitors [99]. This happens through the reprogramming of the kinome that represents a mechanism of drug resistance in response to kinase inhibitors. Therefore, the degradation approach with PROTACs can potentially target all of these aspects avoiding the emergence of drug resistance. Unfortunately, very recently, mechanisms of resistance to bromodomain and extra-terminal domain (BET) proteins degraders have been discovered [100]. The inability of the compounds to degrade the target protein is due to the loss of function of the cullin-RING ligase (CRL) complex specific for each PROTAC, i.e., CRBN- or VHL-related, and can be overcome by sequential administration of the two classes of degraders [100].

Of note, FAK has been found activated or up-regulated in several pediatric cancers, such as renal cancer, neuroblastoma, ewing sarcoma, rhabdomyosarcoma and hepatocellular carcinoma, and FAK depletion or pharmacologic inhibition has resulted in tumor growth impairment, both in vitro and in vivo [4,9,22,32,101,102,103,104,105]. Moreover, FAK expression has been correlated to a worst prognosis in neuroblastoma patients [106]. In a pre-clinical trial on several types of pediatric solid cancer cell lines, such as rhabdomyosarcoma, medulloblastoma, ependymoma, glioblastoma, neuroblastoma and osteosarcoma, VS-4718 (PND-1186) showed limited activity, with event-free survival distribution significantly different compared with controls but with a small shift in magnitude [69]. At present, patients with pediatric cancers are not included in clinical trials with FAK inhibitors. However, the multikinase inhibitor Dasatinib (BMS-354825, SPRYCEL), which targets FAK together with Abl, Src and c-Kit, is under evaluation for rhabdomyosarcoma, neuroblastoma and ewing sarcoma [12].

In summary, identifying the best conditions and drugs that can reduce and/or block FAK functions still remains a challenge. Future investigations should yield novel insights on the knowledge of mechanisms regulated by FAK that can help to develop novel, more selective inhibitors. Moreover, the identification of the best combinations for clinical approaches could also be helpful. Finally, pediatric cancers should be also evaluated to clarify the role of FAK in disease progression and to define an anti-FAK strategy for young patients.

## Figures and Tables

**Figure 1 ijms-23-01998-f001:**
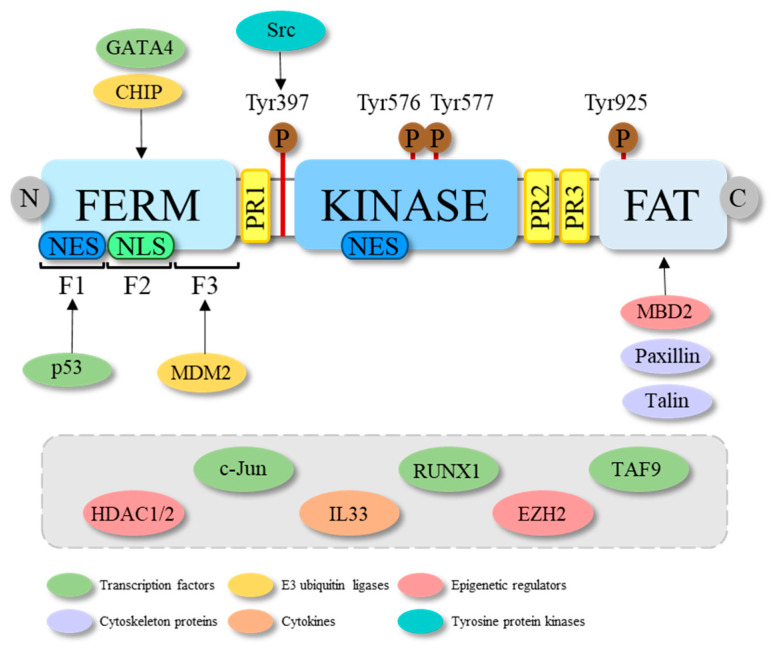
Schematic representation of the molecular structure of FAK and some of its interactors. FAK structure consists of three main domains: the N-terminal FERM, the central kinase and the C-terminal FAT domain. The FERM domain includes three lobes, i.e., F1, F2 and F3, which are bound from transcription factors (among which are GATA4 and p53) and E3 ubiquitin ligases (among which are CHIP and MDM2). While it is known that p53 and MDM2, respectively, bind the F1 and F3 lobes of the FERM of FAK, for GATA4 and CHIP it is, only known that they bind the FERM domain, but unknown is to which of its three lobes. In the FERM domain a nuclear export signal (NES) in the lobe F1 and a nuclear localization signal (NLS) in the lobe F2, responsible for the nuclear export and localization of the protein, are reported. A NES is also found in the kinase domain. FAK structure further comprises three proline-rich (PR) regions that serve as binding sites for the Src homology (SH) 3 domains of several proteins. The main phosphorylation sites are shown with brown circles. In particular, the Tyr397 activation site, the Tyr576 and Tyr577 in the activation loop and the Tyr925 binding site for the SH2 domains are reported. The arrows indicate the FAK domain to which some of its interactors bind, while the grey rectangle shows the FAK interactors with still unknown binding sites.

**Figure 2 ijms-23-01998-f002:**
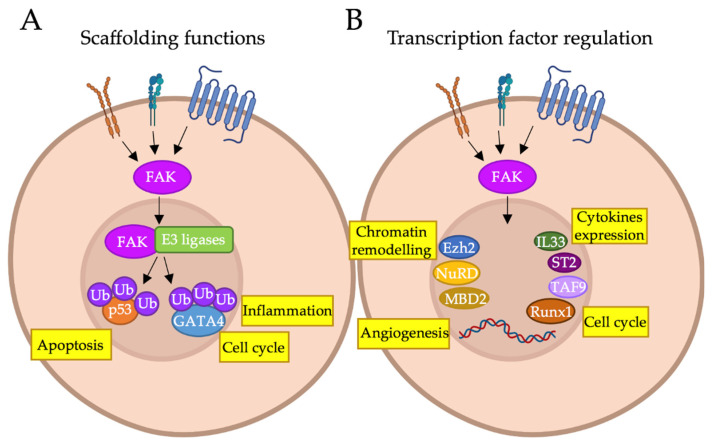
FAK functions in the nucleus. Graphical representation of the nuclear functions of FAK and its interactors. (**A**) FAK acts as a scaffold to stabilize complexes between p53 or GATA4 and the ubiquitin E3 ligases Mdm2 or CHIP, respectively. This leads to p53 and GATA4 polyubiquitination and subsequent degradation by the proteasome; (**B**) FAK acts as a regulator of gene expression by controlling chromatin accessibility, acting on a variety of epigenetic modulators (MBD2, Ezh2, NuRD complex); transcriptionally regulating the expression of transcription factors, such as IL33 and ST2; or forming molecular complexes with transcription factors such as TAF9 and Runx1. By regulating gene expression, nuclear FAK can be involved in apoptosis, inflammation, cell cycle, angiogenesis and cytokine expression. Figure realized with BioRender software (https://biorender.com/, accessed on 27 December 2021).

**Table 1 ijms-23-01998-t001:** Summary of FAK inhibitors in clinical trials.

Drug (Code Name), Trade Name	Target (IC_50_)	Clinical Trial Studies (a, b)	No. of Clinical Trials (a)	Phase (a)
GSK2256098 (GTPL7939)	FAK (0.4 nM)	adenocarcinoma, adult healthy subject, intracranial and recurrent meningioma mesothelioma, pancreatic cancer, pulmonary arterial hypertension, solid cancer	6; 4 completed, 2 active	Phase I: 4 Phase II: 2
VS-4718 (PND-1186)	FAK (1.5 nM)	relapsed or refractory AML, relapsed or refractory B-Cell ALL, metastatic cancer, non-hematologic cancers, pancreatic cancer	3; 2 terminated, 1 withdrawn	Phase I: 3
Defactinib (PF04554878, VS-6063)	FAK (0.6 nM) Pyk2 (0.6 nM)	advanced solid cancer, lung cancer, relapsed malignant and pleural mesothelioma, non-hematologic cancers, NSCLC, ovarian cancer, pancreatic cancer, PDAC	21; 4 terminated, 6 completed 10 active 1 withdrawn	Phase I: 9 Phase I/II: 2 Phase II: 10
VS-6062 (PF00562271)	FAK (1.5 nM) Pyk2 (14 nM)	head and neck cancer, pancreatic cancer, prostatic cancer	1; 1 completed	Phase I: 1
CEP-37440	FAK (2.3 nM) ALK (120 nM)	advanced or metastatic solid tumors	1; 1 completed	Phase I: 1
BI-853520 (IN10018)	FAK	colorectal cancer, metastatic melanoma, metastatic non hematologic malignancies, soft tissue sarcoma, stomach cancer	3; 2 completed, 1 active	Phase I: 3

(a) From www.clinicaltrials.gov (accessed 18 October 2021); (b) AML—acute myeloid leukemia; ALL—acute lymphocytic leukemia; NSCLC—non-small-cell lung carcinoma; PDAC—pancreatic ductal adenocarcinoma.

## Data Availability

Not Applicable.

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
