# Peer review of "New Insights on the Nuclear Functions and Targeting of FAK in Cancer"

_ijms, 2022, doi:10.3390/ijms23041998_

Round 1

Reviewer 1 Report

  1. Kindly include a structure of FAK as Figure 1.
  2. The conclusion section should be re-written.

Author Response

The revised manuscript has been submitted with tracking changes in red.

The English language has been thoroughly checked.

The responses to the Reviewers’ concerns are in red.

REVIEWER #1

  1. Kindly include a structure of FAK as Figure 1

As requested, we have added a Figure 1 reporting the structure of FAK

  1. The conclusion section should be re-written.

The section has been renamed “Future Directions and Concluding Remarks”. In the section, we reported some remarks about the potentiality of anti-FAK strategies in the clinical setting, presenting the pros and cons of each type of approach, including the degraders, and potential future directions. Moreover, we highlighted the insights on FAK activation in pediatric cancers, in which FAK is still understudied.

However, to try to revise the manuscript accordingly to the Reviewer’s comment, we have shuttled two sentences from the “FAK inhibitors” section to the “Concluding Remarks” section to include a discussion on the small molecules targeting the scaffolding functions of FAK in the latter (Page 10, lines 37-41).

Moreover, as for the Reviewer 3 concern, we have added a sentence regarding the importance of studies on the functions of PYK2, the other member of FAK family, in cells inhibited for FAK in determining the choice of the anti-FAK approach in clinic (Pages 10-11, lines 442-445).

Reviewer 2 Report

The authors provide an interesting review on FAK structure and function (scaffolding and transcription factor regulator) as well as different categories of FAK inhibitor according protein structure.

Minors comments: 

signalling or signaling, both are correct however keep the same orthography in the manuscript. (line 3865 74..)

Same remark with number:  thirty two (line 34) vs arabic numerals... 62 patients ( line 240), 34 patients (lines 244) , 47 human (line 253) ……

Table 1:  I wil in the second column the category of FAK inhibitor (Kinase domain inhibitor, allosteric inhibitor or PROTACS) which are well described in the text. I am not sure that “intervention” column adds any value….

Author Response

The revised manuscript has been submitted with tracking changes in red.

The English language has been thoroughly checked.

The responses to the Reviewers’ concerns are in red.

REVIEWER #2

The authors provide an interesting review on FAK structure and function (scaffolding and transcription factor regulator) as well as different categories of FAK inhibitor according protein structure.

Minors comments: 

signalling or signaling, both are correct however keep the same orthography in the manuscript. (line 3865 74..)

We have checked and revised the manuscript accordingly

Same remark with number:  thirty two (line 34) vs arabic numerals... 62 patients ( line 240), 34 patients (lines 244) , 47 human (line 253) ……

We have checked and revised the manuscript accordingly 

Table 1:  I wil in the second column the category of FAK inhibitor (Kinase domain inhibitor, allosteric inhibitor or PROTACS) which are well described in the text.

In the sub-heading “FAK inhibitors”, there is the following sentence: “Indeed, all the compounds inhibiting FAK that have accessed clinical trials and are currently under evaluation in Phase I and II belong to this category … (Table 1).” (Page 6, lines 228-230). Therefore, we think an additional column redundant. However, if the Reviewer believe it is necessary, we will add it.

I am not sure that “intervention” column adds any value….

We have eliminated the column “Interventions” as suggested.

Reviewer 3 Report

This timely review addresses FAK pharmacological inhibition, succinctly describing the structure of the protein, its adhesive (too brief, see below) and nuclear function, and the mechanism of action of different inhibitory approaches. Also included, the state of the art regarding the use of FAK inhibitors in clinical trials to treat cancer.

The manuscript is potentially important and interesting. However, as many of these types of the reviews, it will not stand the test of time particularly well as it will become outdated as clinical trials come and go. However, it can be published if the authors address the following:

  • The review is tremendously skewed by the authors’ viewpoint on the role of FAK. It contains even inflammatory sentences such as in line 41 “However, this concept (FAK was long considered as the central transducer of the integrins’ signaling) was dropped…”. This has not been “dropped”; the role of FAK in the nucleus has been added to the state of the art, without prejudice to the central role of this kinase in adhesive dynamics. The authors need to curb their enthusiasm for their own vision of the field. A cursory search in Pubmed (FAK AND nucleus vs. FAK AND adhesion/integrin) returns 5-10 times more results in the case of adhesion/integrin than nucleus.
  • The differences in terms of nuclear signaling between FAK and Pyk2 need to be discussed.
  • This can be found elsewhere, but the authors need to include a FAK domain breakdown figure, since a lot of their discussion focuses on the description of different domains inducing different functions. The authors can add to the field by including the potential nuclear interactions in the same figure.
  • The functions of FAK in the nucleus are described, but poorly integrated. Data pertaining to the control of VCAM-1 expression (which would be an inflammatory function) are thrown together with data on acetylation and p53 interactions. This needs to be re-conceived and ordered in a form that is logical and coherent to the reader. This is particularly important as this is the least explored part of the FAK canon.
  • The section on the mechanism of inhibitors is okay, but this reviewer lacks a thorough description of the observed effects of each inhibitor in the context of adhesion/integrin signaling vs. nuclear regulation in vitro and in vivo (away from clinical trials).

Author Response

The revised manuscript has been submitted with tracking changes in red.

The English language has been thoroughly checked.

The responses to the Reviewers’ concerns are in red.

REVIEWER #3

This timely review addresses FAK pharmacological inhibition, succinctly describing the structure of the protein, its adhesive (too brief, see below) and nuclear function, and the mechanism of action of different inhibitory approaches. Also included, the state of the art regarding the use of FAK inhibitors in clinical trials to treat cancer.

The manuscript is potentially important and interesting. However, as many of these types of the reviews, it will not stand the test of time particularly well as it will become outdated as clinical trials come and go. However, it can be published if the authors address the following:

  • The review is tremendously skewed by the authors’ viewpoint on the role of FAK. It contains even inflammatory sentences such as in line 41 “However, this concept (FAK was long considered as the central transducer of the integrins’ signaling) was dropped…”. This has not been “dropped”; the role of FAK in the nucleus has been added to the state of the art, without prejudice to the central role of this kinase in adhesive dynamics. The authors need to curb their enthusiasm for their own vision of the field. A cursory search in Pubmed (FAK AND nucleus vs. FAK AND adhesion/integrin) returns 5-10 times more results in the case of adhesion/integrin than nucleus.

We apologize for the inaccuracy in presenting the “nuclear” aspect of FAK. The sentence has been rephrased: “Over the last ten years, in addition to this kinase-dependent function, an unexpected kinase-independent role in scaffolding and regulation of gene transcription was also discovered for FAK ….” (Page 1, lines 42-45).

Moreover, we have highlighted in the Abstract section (Page 1, lines 22-23) and in the Introduction section (Page 2, lines 54-56) that our review is mainly focused on the novel insights into the scaffolding functions of FAK.

We have also revised the Title accordingly.

  • The differences in terms of nuclear signaling between FAK and Pyk2 need to be discussed.

We have revised the “FAK inhibitors” section adding some sentences about the “…challenges in the development of FAK inhibitors … to achieve …. selectivity towards the ….PYK2…” and on some major similarities and differences between the two proteins (Page 6, lines 230-244). However, since the review is focused on FAK, to avoid confusion we have only highlighted the similarity in the p53 regulation and MDB2 binding, and the difference in the response to Calcium by PYK2. Moreover, as for the Reviewer 3 concern, we have added a sentence regarding the importance of studies on the functions of PYK2, the other member of FAK family, in cells inhibited for FAK in determining the choice of the anti-FAK approach in clinic (Pages 10-11, lines 442-445).

  • This can be found elsewhere, but the authors need to include a FAK domain breakdown figure, since a lot of their discussion focuses on the description of different domains inducing different functions. The authors can add to the field by including the potential nuclear interactions in the same figure.

A novel Figure 1 has been added to the manuscript accordingly with the Reviewer’s concerns.

  • The functions of FAK in the nucleus are described, but poorly integrated. Data pertaining to the control of VCAM-1 expression (which would be an inflammatory function) are thrown together with data on acetylation and p53 interactions. This needs to be re-conceived and ordered in a form that is logical and coherent to the reader. This is particularly important as this is the least explored part of the FAK canon.

The section has been completely revised accordingly to the Reviewer’s concerns. Moreover, we changed the title of the subheading as following “Nuclear FAK Functions Related to Cancer” to focus on those FAK nuclear features more connected to our review aim.

  • The section on the mechanism of inhibitors is okay, but this reviewer lacks a thorough description of the observed effects of each inhibitor in the context of adhesion/integrin signaling vs. nuclear regulation in vitro and in vivo (away from clinical trials).

To revise the section accordingly to the Reviewer’s comment, we have added more results about the use of the reported inhibitors in preclinical works. We found two references on VS-4718 (PND-1186) and Defactinib highlighting the evidence of nuclear FAK under treatment (Refs 72 and 77).

Round 2

Reviewer 3 Report

The authors have addressed my concerns seriously and in a scientifically accurate manner. Well done!